# Corrosion Behavior of Alumina-Forming Austenitic Steel in Supercritical Carbon Dioxide Conditions: Effects of Nb Content and Temperature

**DOI:** 10.3390/ma16114081

**Published:** 2023-05-30

**Authors:** Zhaodandan Ma, Shuo Cong, Huan Chen, Zhu Liu, Yuanyuan Dong, Rui Tang, Tian Qiu, Yong Chen, Xianglong Guo

**Affiliations:** 1Science and Technology on Reactor System Design Technology Laboratory, Chengdu 610041, China; npicmzdd@163.com (Z.M.); sshuimus@163.com (Y.D.); quiter777@163.com (T.Q.); 2Science and Technology on Reactor Fuel and Materials Laboratory, Chengdu 610041, China; npicchh@163.com (H.C.); xajttr@163.com (R.T.); 3Nuclear Power Institute of China, Chengdu 610041, China; 4School of Nuclear Science and Engineering, Shanghai Jiao Tong University, Shanghai 200030, China; congshuo@sjtu.edu.cn (S.C.); zhuzh_2010@sjtu.edu.cn (Z.L.)

**Keywords:** alumina-forming austenitic (AFA) stainless steel, supercritical carbon dioxide, high-temperature corrosion, different Nb additions, SEM, TEM

## Abstract

The corrosion behavior of alumina-forming austenitic (AFA) stainless steels with different Nb additions in a supercritical carbon dioxide environment at 500 °C, 600 °C, and 20 MPa was investigated. The steels with low Nb content were found to have a novel structure with a double oxide as an outer Cr_2_O_3_ oxide film and an inner Al_2_O_3_ oxide layer with discontinuous Fe-rich spinels on the outer surface and a transition layer consisting of Cr spinels and γ’-Ni_3_Al phases randomly distributed under the oxide layer. Oxidation resistance was improved by accelerating diffusion through refined grain boundaries after the addition of 0.6 wt.% Nb. However, the corrosion resistance decreased significantly at higher Nb content due to the formation of continuous thick outer Fe-rich nodules on the surface and an internal oxide zone, and Fe_2_(Mo, Nb) laves phases were also detected, which prevented the outward diffusion of Al ions and promoted the formation of cracks within the oxide layer, resulting in unfavorable effects on oxidation. After exposure at 500 °C, fewer spinels and thinner oxide scales were found. The specific mechanism was discussed.

## 1. Introduction

The development of a nuclear reactor cooled with supercritical carbon dioxide (S-CO_2_) has become an important topic in recent years and has attracted much attention [1,2]. Compared with the traditional steam cycle and conventional reactor, S-CO_2_ has the advantages of high energy conversion efficiency, centralized piping, high space utilization, and better safety [3]. In S-CO_2_ systems, the main components are directly exposed to high operating temperature and high gas pressure, which is the main problem because an extreme situation greatly increases the risk of degradation phenomena such as property damage, loss of structural integrity, and failure [4,5]. In the above damage, the corrosion problem of the construction materials is the main problem, and the corrosion products generated during operation significantly deteriorate the performance of the construction material [6]. Therefore, it is important to investigate the corrosion behavior of the materials under consideration in the S-CO_2_ system.

Various materials have been used, ranging from ferritic–martensitic (F-M) stainless steel to austenitic stainless steel and Ni-based superalloys [4,5,6,7,8,9,10]. Researchers all over the have carried out research on the corrosion of these materials in the S-CO_2_ environment and obtained some useful information [4,5,6,7,8,9,10]. However, the incompatibility of mechanical properties, corrosion problems, and cost of materials limit the applications [4,5,6,7,8,9,10].

Recently, an alumina-forming austenitic steel (AFA) was developed by Oak Ridge National Laboratory, which exhibits good oxidation resistance with Al_2_O_3_ formation on the surface and excellent creep properties with nanoscale precipitates [11]. Previous studies have shown that high chromium content can effectively improve corrosion resistance due to the stable Cr_2_O_3_ oxide layer, while the Al_2_O_3_ scale had lower growth rates and better thermodynamic stability compared to the Cr_2_O_3_ scale, which was found to be protective under complicated nuclear systems [12]. There have been many studies on the corrosion resistance of AFA steel in the S-CO_2_ environment [13,14]. Pint et al. [13] studied various structural materials under different pressures and found that the corrosion resistance of AFA steel was even better than that of Ni-based alloys with special components, which was mainly due to the dense Al_2_O_3_ oxide layer. He et al. [14] analyzed the structure of the oxide film of AFA steel after corrosion in S-CO_2_ and confirmed a multilayer structure composed of various spinels and a compact Al_2_O_3_ layer, which can further improve the corrosion ability in extreme environments.

In fact, the alumina content is the key component for the formation of the Al-rich oxide layer. The Nb content is as important as the Al element for α-Al_2_O_3_ [15]. Previous studies have shown that higher Nb content than Al is required for the protective layer [15,16]. Yamamoto [15] confirmed that the increased Nb content (between 0.6 and 1 wt.%) was suspected to be a key factor for the formation of the external oxide layer. In addition, AFA-Nb steel exhibited improved oxidation resistance after oxidation at 1173 K in air, which was attributed to the formation of the B2-NiAl phase by Nb addition from the studies of Brady [17]. B2-NiAl precipitates usually were identified as reservoirs for maintaining a stable Al_2_O_3_ layer in AFA steel [18,19]. However, with further increasing Nb content, the oxidation resistance decreased in these studies. Shen [20] showed that higher Nb content deteriorates oxidation resistance, and many Fe_2_Nb laves formed may have complex impacts on oxidation performance after the addition of 4.5 wt.% Nb content. To date, no specific mechanistic understanding of the effects of Nb on oxidation has been achieved.

Therefore, the aim of this study was to investigate the corrosion problem of AFA steels with different Nb contents under S-CO_2_ conditions at 500 °C, 600 °C, and 20 MPa. The measurement of weight gain and microstructure was carried out to evaluate the corrosion behavior of AFA steels. We also analyze the effects of Nb content and temperature on corrosion performance and discuss the mechanism. These results may provide a necessary reference for the application of AFA steels in advanced S-CO_2_ energy systems.

## 2. Material and Experimental Methods

### 2.1. Material Preparation

The chemical composition of the three AFA steels was prepared by vacuum melting and is summarized in Table 1. From this, it can be seen that these AFA steels have the same composition but different Nb contents (0~2.5 wt.%), and all of them are austenitic steels. The actual values of composition measured by X-ray fluorescence (XRF) are also shown in Table 1. To refine the grains and remove impurities generated during the melting process, AFA steels must be homogenized and forged. To avoid the influence of the original and coarsened precipitates on the relative corrosion behavior, solid solution treatment was performed at 1200 °C for 1 h.

### 2.2. Experimental Methods

Isothermal corrosion experiments on AFA steels were carried out in this work at Shanghai Jiaotong University, using an S-CO_2_ corrosion system, as shown in Figure 1a. The experimental system is mainly divided into two separate facilities: the gas fluid control loop and the corrosion autoclave. High-purity CO_2_ (mass fraction > 99.995%) was introduced into the autoclave by a high-pressure pump, and the circulating gas was finally collected. The autoclave was prepared by Ni-based alloy 625 (self-made), which can reach a high operating temperature of nearly 700 °C and 25 MPa to ensure the accuracy and stability of the results.

During the corrosion test, the S-CO_2_ mass flow rate was maintained at 0.5 kg/h, and the corrosion conditions were conducted at 500 °C, 600 °C, and 20 MPa. The coupon-like specimens had a diameter of 10 mm and a thickness of 2 mm. A 2 mm hole was drilled at the top of the specimen for hanging in the autoclave. An alumina boat was suspended from the specimens with platinum wire, and alumina spacers were attached to prevent direct contact with each other. To avoid the effects of scratches on corrosion performance, the specimens were polished. The surface of the specimens was first polished with SiC sandpaper and then finely ground with a polishing cloth with the use of an automatic polishing machine. Finally, the specimens were washed with alcohol and ultrapure water and dried. The specimens were removed from the autoclave according to the corrosion time of 100 h, 300 h, 500 h, and 1000 h. For each type of alloy and each test condition, three specimens were tested.

For each period of corrosion testing, the weight gain of each sample was measured using an electronic microbalance with an accuracy of 0.001 mg. A Brucker instrument (Billerica, MA, USA) (D8 QUEST ECO) was used to detect the oxide phases formed on the material surface and precipitates in the matrix in the diffraction range of 20°~80° at a rate of 0.02 step/s. The surface microstructures of the corroded samples were characterized using the FEI Magellan 400 field emission scanning electron microscope (FE-SEM, Apero 2C) (Thermo Fisher Scientific, Waltham, MA, USA). In addition, a transmission electron microscope (TEM, Talos F200X, Thermo Fisher Scientific, Waltham, MA, USA) was used to characterize the microstructure of the cross-section of the oxide layer. An energy-dispersive spectrometer (EDS, 20 kV 15 Bi) with TEM was used to study the composition and distribution of elements in the cross-sectional area. Selected area electron diffraction (SAED) was used to analyze the precipitates under the oxide. The focused ion beam method (FIB) was used to prepare the oxide samples for TEM testing.

In the analysis and discussion section, we consulted literature to describe the reactions during the corrosion process and used drawing tools such as PowerPoint to abstractly depict the corrosion mechanism.

## 3. Results

### 3.1. Microstructure before Corrosion

The distribution of precipitates in the samples before corrosion is very important for the corrosion properties. Figure 2 shows the distribution and change of precipitates in the matrix before corrosion. In Figure 2a, almost no precipitates were found in the matrix of A1-0Nb alloy, but in Figure 2b,c, a few white precipitates with a size up to 2 μm were distributed at the grain boundary. EDS Spectrum proved that the coarse precipitates were NbC carbides. The primary NbC precipitates usually form during the solidification phase and are difficult to dissolve after solution treatment at 1200 °C. Nb-containing laves phases also form readily at high temperature. Nano-sized precipitates formed in the matrix of A2-0.6Nb steel were observed by TEM in Figure 2d,e, which was calibrated as nano-sized NbC. It is certain that with increasing Nb content, the number of Nb-containing precipitates gradually increases.

### 3.2. Weight Gain and Corrosion Kinetics

Figure 3a shows the weight gain of three AFA steels after being exposed to the S-CO_2_ environment at 600 °C and 20 MPa for up to 1000 h. The three AFA steels all obeyed the law of parabolic growth rate or approximately a parabolic trend, which is typical for the presence of a compact and protective oxide layer in S CO_2_ [5,6,7]. The weight changes of A2-0.6Nb steel were minimal and maintained a low corrosion rate, while for A1-0Nb and A3-2.5Nb steels, they both behaved with an abrupt trend of weight gain after 700 h. This particular transition in rates is a result of the rapid increase in Fe-rich nodules. Compared with the previous studies, AFA steels have a significantly lower weight gain and corrosion rate during the corrosion process, even with differences of an order of magnitude, indicating that AFA steel has excellent corrosion resistance in S-CO_2_ [21].

Of the three steels, the corrosion mass of A3-2.5Nb steel exhibited the greatest weight gain (16.3 mg/dm^2^) after 1000 h of exposure, while A2-0.6Nb steel had the best corrosion resistance with a weight gain of only 2.11 mg/dm^2^, which was even better than A1-0Nb steel. With the same content of other elements, the trend of corrosion weight increase is closely related to the content of Nb additions. The higher or lower the Nb content, the greater the corrosion mass increase, and the poorer the corrosion resistance. To understand the difference behind the observed weight gain behavior, further analysis of the oxide formed on the specimen surface must be performed, which is described in the following sections.

Figure 3b compares the corrosion kinetic curves of A2-0.6Nb steel at 500 °C and 600 °C with different corrosion times. As the corrosion temperature increased, the corrosion weight increased significantly. The kinetic curves at 500 °C fluctuated greatly, which was less than the values of 600 °C all the time, but higher when exposed at 700 h. The sudden increase may be caused by the different surface conditions of the processed samples [22]. This phenomenon will be explained in detail later.

### 3.3. Images of Corrosion Sample

Figure 4 shows the photos of AFA steels after exposure to S-CO_2_ at 500 °C and 600 °C for 1000 h. At 600 °C, a light blue color was observed on the surface of A2-0.6Nb steel, possibly due to a thin oxide layer. When no Nb or 2.5 wt.% Nb was added, the surface color gradually changed to yellow and brown, especially at the edges. Corners and holes in A1-0Nb steels indicated relatively severe and uniform corrosion. When Nb content increased to 2.5 wt.% in Figure 4c, the surface became completely dark brown, which may be caused by increasing corrosion products. In Figure 4d, the surface remained light blue after corrosion at 500 °C, but some dark spots appeared on the surface.

### 3.4. Characterization of Oxidation

#### 3.4.1. Surface Morphologies of the Oxide Films

Figure 5 shows the surface morphology SEM of the oxide layers of the three steels exposed to 600 °C S-CO_2_ for 500 h and 1000 h. The oxide layer was fairly uniform, but some nodules were found. A2-0.6Nb steel exhibited a relatively loose and compact structure of the oxide layer, and the nodules were quite small (inset in Figure 4d). In A1-0Nb steel, on the other hand, the nodules became larger and were more clustered (see Figure 4a,b). On the surface of A3-2.5Nb steel, even larger multi-edged iron oxide grains were detected after corrosion at 600 °C, and the size of iron oxide particles increased with the extension of corrosion time (see Figure 4c). This result is consistent with the weight increase due to corrosion. A2-0.6Nb steel showed the best corrosion resistance with a stable oxide structure. It was generally believed that the transition of weight increase in Figure 3a of steels is related to large oxide nodules in the microstructure morphology. Normally, a rapid increase in Fe-rich or Nb-rich oxides can easily lead to an abrupt transition of weight increase [23]. The changes in A3-2.5Nb steel were quite distinct, showing a clear picture of coarse oxide clusters and an almost exponentially growing weight gain curve.

It can be found that the oxide layer of 500 °C had fewer oxidized nodules than the surface morphology of the oxide layer of 600 °C. However, there were some scratches on the surface after 500 °C exposures in Figure 5g,h, which may be caused by uneven grinding during preparation. Additionally, in Figure 4d, we can see that there are many dark brown spot products generated due to uneven surface conditions. According to the literature, uneven surfaces will have an obvious impact on corrosion performance, especially in the later stages of corrosion, which also explains why the weight gain fluctuates significantly after 500 h [22,23].

#### 3.4.2. Precipitate and Oxide Film after Corrosion

Figure 6 shows the results of the XRD analysis of the oxide layer and matrix of the three steels after corrosion. In Figure 6a, a small amount of FeCr_2_O_4_ peaks appears after corrosion, while large matrix peaks (α-Fe) can be seen in the XRD pattern, indicating a thin oxide layer and the formation of fewer corrosion products. Furthermore, a large amount of FeCr_2_O_4_ was detected in A3-2.5Nb steel. After corrosion, the diffraction peaks of γ’-Ni_3_Al, NbC carbides, B2-NiAl phase, and laves phase can be clearly seen in Figure 6b.

#### 3.4.3. Detail Microstructures of the Oxide Film

The contrast of the oxide layer in the cross-section of the three AFA steels with different Nb contents detected by SEM is shown in Figure 7. The scale thickness generally changed with Nb content. The oxide layer was not continuous in A1-0Nb steel, but a thick oxide structure appeared in A3-2.5Nb steel.

To clarify the mechanism of AFA corroded in S-CO_2_, the cross-sectional micrographs and EDS mapping results of TEM of A2-0.6Nb steel after corrosion in S-CO_2_ at 600 °C for 1000 h were studied, and the structure of the oxide film was analyzed. Figure 8 shows the morphology of the oxide film in the cross-section of A2-0.6Nb steel with the best corrosion performance. The thickness of the oxide film was about 100 nm, measured by the EDS line spectrum after 1000 h of exposure. Combined with the HAADF image and mapping spectrum, the section of the oxide layer of A2-0.6Nb steel showed the typical double-oxide layer. The outer oxide layer was composed of Cr-O with a small amount of Fe, and the inner layer was mainly composed of Al-O oxide products. According to previous studies, the Fe-Cr oxide distributed in the outermost discontinuous layer is generally FeCr_2_O_4_ spinels [24,25]. Due to the scarcity of Fe element in this study, the quantity of spinels is limited. XRD results can also confirm that there are some FeCr_2_O_4_ spinels produced after corrosion. The spinels could be consistent with the tiny nodules on the oxide layer in Figure 5d. It is worth noting that there were some discontinuous Cr-Al spinels in the Cr oxide layer. Except for FeCr_2_O_4_ spinels, the SAED result confirmed that there was also a separated Cr_2_O_3_ layer presented in the outer layer, and the inner layer was a dense and compact Al_2_O_3_ oxide film based on the data. Therefore, it can be concluded that after corrosion in S-CO_2_ for 1000 h, the double layer was composed of a continuous Cr_2_O_3_ layer with dispersed spinels, and the inner layer was fully Al_2_O_3_ in AFA steel.

In the matrix under the oxide layer, there were a small number of precipitates, mainly divided into two types. One consisted of Cr-Mo-C-O elements, which were found to be Cr-rich spinels with a size of 250 nm. This carbon distribution could be due to the high Cr content in the matrix. The other was compounds of Ni and Al, which were identified to be γ’-Ni_3_Al with a polygonal shape and a size of 200 nm.

Below the inner oxide layer, there was a region where Al and Cr ions were very rare. This region is referred to as the Al- or Cr-denuded zone in prediction studies [15]. As shown in Figure 8a, the Al-denuded zone of A2-0.6Nb steel is about 160 nm after rough measurement. In this region, Al and Cr ions have diffused outward and formed oxides, resulting in a low content of Al or Cr elements. Pint [26] proposed a study of an oxidation test with 10% water vapor at 1073 K after an exposure time of 15,000 h and demonstrated the existence of an Al-denuded zone in AFA steel for the first time, which reflects the degree of oxidation. In another study by Wen [27], an Al-denuded zone with a thickness of up to 4.5 μm appeared after high-temperature oxidation. The alloy undergoes an intense oxidation reaction, resulting in the formation of an oxide film with a thickness of 1 μm. That means the higher the degree of oxidation, the more Al atoms need to be consumed, and the length of the Al- or Cr-denuded zone is larger.

To analyze the effect of Nb on corrosion oxide film in detail, the cross-sectional micrographs and EDS mapping results observed by TEM of A1-0Nb and A3-2.5Nb steels after corrosion in S-CO_2_ at 600 °C for 1000 h ere investigated in this section, and the structure of the oxide film was analyzed to compare with A2-0.6Nb steel above. The structure of the oxide film of A1-0Nb steel is shown in Figure 9. After S-CO_2_ exposure, the oxide film remained as a double-oxide film as well, which is displayed as a black and dark layer in the HAADF image. Based on the data of EDS mapping in Figure 9b, the outer layer was an Fe-Cr oxide layer, and the inner layer was composed of Al and O, which show the same structure as A2-0.6Nb steel. Similarly, we can conclude that the oxide layer consisted of FeCr_2_O_4_ spinels outmost the layer, and a Cr_2_O_3_ and Al_2_O_3_ double-oxide layer was formed in the external area. Furthermore, Al element was inserted in the Cr oxide layer, indicating the formation of a small amount of Cr-Al spinels.

Below the oxide film, Cr-rich precipitates and Ni-Al compounds also appeared in A1-0Nb steel, which were identified as Cr-rich spinels and γ’-Ni_3_Al phases, as well as A2-0.6Nb steel. The size of Ni-Al compounds in A1-0Nb steel was less than 100 nm, but the number of precipitates increased significantly compared with A2-0.6Nb steel, even at larger multiples. Meanwhile, the Al- and Cr-denuded zone was about 200 nm here after measurement, which was larger than that of A2-0.6Nb steel. As mentioned above, the distance length of the Al- and Cr-denuded zone is related to the degree of oxidation. In other words, more atoms were absorbed for oxidation in A-0Nb steel, and a more intense oxidation reaction took place.

As the most corroded of the three AFA steels, the cross-sectional oxide of A3-2.5Nb steel was examined by TEM, and the results are shown in Figure 10. When the amount of Nb additions increased to a high level, the microstructures of the oxide changed slightly compared to the other two steels. EDS showed that the outer oxide layer was mainly composed of Fe, Cr, and O elements, which were confirmed to be FeCr_2_O_4_. It did not undergo the same changes as the other two AFA steels, and only the Fe oxide layer was more pronounced. What is more, the oxide layers were not as dense as steels with low Nb content, but pores and cracks were found in the outer layer or at the interface between the two oxide layers.

Different types of precipitates were discovered in the matrix under the oxide scales. Some internal Al_2_O_3_ oxide and Cr oxide products were observed under the matrix/oxide interface, suggesting that the unconsolidated Fe-Cr oxide layer relatively easily allows the diffusion of O elements inward and accelerates an internal oxide zone (IOZ) in the final state. In addition, the precipitates of Mo, Nb, and Fe elements also appeared in the matrix near the oxide layers, which was certified as the Fe_2_(Mo, Nb) laves phase. The laves phase sometimes is believed to have a negative effect on oxidation, leading to the deterioration of oxidation resistance in many studies [28,29]. The Al- and Cr-depleted zone was about 300 nm in size, which means more significant oxidation.

To determine the oxide structure and precipitates after corrosion at 500 °C, the microscopic detailed images are shown in Figure 11. Double-oxide layers still remained, but almost no precipitates of Cr were observed at low temperature, only several Ni-Al phases. The Al- and Cr-depleted zone corroded at 500 °C dwindled to only 80 nm. According to the Arrhenius formula, the diffusion rate becomes faster with improving temperature [30]. Therefore, there are more Fe-Cr- or Cr-Al-rich spinels in the oxide layer at 600 °C. Additionally, the growing number of oxide and precipitates was mainly caused by the accelerated diffusion of ions. As a result, precipitates cannot be observed clearly at 500 °C because of the low diffusion rate of ions [31].

Figure 12 shows the comparison of cross-sectional micrographs and EDS mapping the results of A2-0.6Nb steel at 500 °C and 600 °C for 1000 h. The micrographs of the cross-sections of A2-0.6Nb steel showed a thickness of the oxidation layer at high temperature with a distinct Cr_2_O_3_ and Al_2_O_3_ oxide layer with many spinels. However, the continuous oxide layers seem to be unconsolidated. Although the oxide layer after corroding at 500 °C was not as thick as that at high temperature, it exhibited a relatively rock-solid structure.

## 4. Discussion

### 4.1. The Mechanism of AFA in S-CO_2_

Combining Figure 8, Figure 9 and Figure 10, the oxide products of AFA steels consist of Cr_2_O_3_, Al_2_O_3_, and various spinels. At the beginning of the oxidation reaction, the rapid uptake of oxygen converts the surface layer into oxides [14]. The oxidation process in S-CO_2_ can be explained by the mechanism of cavity-induced duplex-oxide formation. Rouillard et al. [32,33] proposed that the kinetics of internal oxide growth during corrosion of 9Cr steel in CO_2_ are controlled by outward iron diffusion, which is called the “available space model”, i.e., ions diffuse outward to form oxides. The external CO_2_ gas molecules diffuse to the oxide/metal interface, react with nearby metal ions, and provide growth space for the newly formed oxide. The “available space” model explains the formation of the double-oxide layer that corrodes in the S-CO_2_ environment of 9-12Cr alloys. The oxidation process is mainly controlled by Fe in the first phase [7,8,9,10,14]. The subsequent Cr_2_O_3_ oxide still allows Fe to diffuse outward, but the dense Cr_2_O_3_ protective layer can effectively reduce the diffusion rate afterward, which controls the next phase in the steels without Al. Yang [30] investigated the oxide layer of T91 and 316 steel after corrosion in S-CO_2_ and found that coarse Fe_3_O_4_ formed on the surface of T91 with low Cr content, which eventually led to an oxide layer as thick as 28.3 μm and IOZ. While in 316 high Cr steel, large Fe-Cr spinels formed in these alloys, only few Fe-O particles were observed on the surface of 316. In the same way, Fe ions also diffuse into the outermost layer in AFA steel containing Al additions, and the diffusion rate of Cr and Al ions is relatively slow compared to Fe [14]. However, due to the quite high Cr content in AFA steel, Cr_2_O_3_ is formed closely and quickly. Thus, the outer individual Fe-Cr spinels formed instead of stable Fe_3_O_4_, even in A3-2.5Nb steel with the worst corrosion resistance.

As for the continuous double-oxide layer of AFA steels, the outer Cr_2_O_3_ layer forms earlier, leaving “free space” for the growth of the inner Al_2_O_3_ oxide. This phenomenon is not only caused by the higher Cr content but is also related to differences in the growth rate of separated oxide layers. Thus, the oxidation rate is initially controlled by the Cr_2_O_3_ oxide film temporarily, but once Al_2_O_3_ is formed, the oxidation reaction rate is mainly influenced by the Al oxide film [34,35]. Reactions (1) and (2) show that Al_2_O_3_ is stable. It is more difficult for external O atoms and internal Fe ions to penetrate from the dense Al_2_O_3_ or double-layer oxide film, which has lower solubility of Fe ions. For the above reasons, only Fe-Cr spinels are formed, and the oxidation reaction rate in AFA steels is greatly reduced. Overall, the excellent oxidation resistance of AFA steel is mainly attributed to the stable Al_2_O_3_ oxide film. Secondly, it is attributed to the Cr_2_O_3_ oxide film, and to be precise, this double layer further enhances the corrosion resistance of the alloys.
2Cr + 3/2CO_2_(g) = Cr_2_O_3_ + 3/2C, ΔG = −309.786 kJ/mol(1)
2Al + 3/2CO_2_(g) = Al_2_O_3_ + 3/2C, ΔG = −808.387 kJ/mol(2)

Below the oxide layer, there are usually Ni-Al precipitates as observed in this paper near the oxide/matrix interface that support the growth of the Al_2_O_3_ layer [26,27]. In previous work, mainly B2-NiAl phases [15,20,26,27] were found near the layer. On the contrary, γ’-Ni_3_Al phases were formed in this work, which was a rare situation. There are several reasons: the first is that the initial grain size of the alloy before corrosion in Figure 1 is relatively large, which may not be conducive to precipitates [36,37]. Second, the γ’-Ni_3_Al phase formed here may be related to the unbalanced ratio of Al and Ni content. According to previous research, when the Ni content exceeds 30 wt.%, it is more likely to precipitate γ’-Ni_3_Al in the thermal process [38,39]. While the Ni content is up to 25 wt.% here, the excess Ni atoms are more easily bonded to Al atoms, especially in the Al-denuded zone. Another reason is the mild oxidation process of S-CO_2_ corrosion. The obvious B2-NiAl phase was found in the study conducted at a temperature of 1073 K in dry air [14,15]. High temperature would promote the transformation of these unstable precipitates into stable structures in most cases [39,40]. In addition to the Ni-Al phase, Cr-rich precipitates appeared at low Nb addition. Despite the C concentration in the matrix being only 0.02 wt.%, the Cr-rich spinels may be caused by CO produced in the oxidation reaction. As a kind of gas molecule, CO penetrates through the double-oxide layer relatively easily.

The carbonization reaction is a notable feature of S-CO_2_ corrosion. Oxidation and carbonization promote or interact with each other [34]. Carbon is usually produced in the processes of oxide formation or comes from the decomposition of carbon dioxide, such as Reaction (3).
2CO(g) → CO_2_ + C(s)(3)

However, according to the surface and cross-section observation of the oxide layer in our work, C deposits did not appear in AFA steel. Instead, carbonization usually occurred in these alloys with poor oxidation protection. Chen [23] investigated four different alloys with different Cr contents and found that low Cr alloys have worse antioxidant performance, which also had a distinct carburization layer after corrosion in S-CO_2_, while only a few carbides formed in these alloys with excellent oxidation protection and better corrosion resistance. This is because as the oxidation reaction rate decreased, the decomposition of CO_2_ or carburization reaction reduced subsequently. Therefore, as displayed in our work, the dense Al_2_O_3_ oxide layer formed in AFA steel inhibited the diffusion of CO and CO_2_ effectively, which further suppressed the carburization phenomenon.

### 4.2. The Effect of Nb on Corrosion Behavior

The oxide film structure of the three steels with different Nb additions was compared for a detailed analysis of the influence of Nb element on corrosion. The comparative data also come from the TEM results. According to rough statistics, the thicknesses of the three steels were 110 nm, 100 nm, and 300 nm, respectively. In general, there was not much difference in the thickness of the oxide film between A1-0Nb and A2-0.6Nb, despite the thickness of the oxide film on A3-2.5Nb steel changing significantly, and the chemical composition of the alloy oxide film was almost the same. The oxide layer structure of the three steels all exhibited a double-structure oxide layer with outer Cr_2_O_3_ and inner Al_2_O_3_. Meanwhile, some Fe-Cr spinels were distributed sporadically outside the oxide layers. Although the structure and chemical composition of the oxide did not undergo fundamental changes, the changes were mainly reflected in the quantity and distribution of corrosion production. For example, compared to A1-0Nb steel, the amount of Cr-Al spinels in A2-0.6Nb steel increased, and the thickness of the Cr oxide film also improved. However, for A3-2.5Nb steel, there were almost no Cr-Al spinels in the outer layer, but rather all inner Al_2_O_3_ grew inward. Although the thickness of Cr_2_O_3_ and Al_2_O_3_ oxide films gradually increased with increasing Nb content, the oxide film became less dense, especially in A3-2.5Nb steels, where the density of the structure decreased significantly with large or small pores. Through previous research, there have been significant changes in the precipitates caused by high-temperature corrosion as well. Laves phases including Nb content gradually occupied the majority when the amount of Nb additions increased.

Previous work has certified that after adding 0.6 wt.%Nb, the corrosion resistance of steel is improved greatly. The main reason is the thick oxide layer and large multi-spinels. First of all, as shown in Figure 2, the addition of Nb resulted in grain refinement and an improvement in the density of grain boundaries. For A1-0Nb steel, the grain size was about 100 microns, while the grain size gradually decreased with increasing Nb content. The grain size decreased to more than ten micrometers with the addition of 2.5 wt.% Nb. It is easier for atoms to diffuse at grain boundaries and form products during corrosion [37]. As for why Nb could refine grains, most studies have explained that the pinning effect of precipitates with Nb components hinders the growth of grain boundaries [41,42].

In addition, a small amount of Nb-Cr or Nb-Al compounds is also helpful for forming the oxide film [43]. This kind of compound was not observed clearly before corrosion in this work. However, it cannot be excluded that there are still some transient intermediate substances that are too small to detect. Moreover, small Nb additions will affect the solubility of other elements in the α-Fe matrix [15]. Brady et al. [44] found that an increasing Nb content decreases the solution limit of Al elements slightly, which leads to the promotion of the formation of the Al_2_O_3_ film, and small Nb addition reduces oxygen as well. Weng [45] demonstrated that Nb can improve oxidation resistance with a decrease in oxygen vacancies and suppress the inward diffusion of oxygen in Ti-Al-Nb steel.

However, as Nb content increases to 2.5 wt.%, the decline in corrosion resistance becomes pronounced, and the reasons for this are more complicated. The comparison of three steel is shown in Figure 13. It is interesting to note why the thickness of the oxide film increases but the corrosion behavior of A3-2.5Nb steel presents opposite trends. As we know, the consecutive oxide film is not only the key to excellent corrosion behavior but also the entire structure of the corrosion product, including oxidation products, carburization products, and precipitates near the oxide film. The most important reason is the sharp increase in the number of Fe oxide spinels, which becomes more prominent as shown in Figure 10b and Figure 13c as Nb content increases. Brady et al. [44] demonstrated that alloys with higher Nb content exhibiting mass loss may be attributable to Fe-rich oxide nodules. Shen et al. [20] also observed the same phenomenon: Fe-rich nodules led to a significant deterioration of antioxidant properties. Of course, the improvement in the number of Fe oxide spinels is also attributed to the refined grains and the promotion of ion diffusion, as depicted in Figure 14. Of course, the improvement in the number of Fe oxide spinels is also attributed to the refined grains and the promotion of ion diffusion.

In this work, the scattered Fe-Cr spinels at low Nb content completely transformed into an almost fully covered Fe-Cr oxide layer after the addition of 2.5 wt.% Nb content. This Fe-Cr oxide often protrudes upward and reaches a size of about 1 μm in A3-2.5Nb steel, represented by the aggregated oxide nodules in the surface morphology in Figure 5f. Additionally, the photos of the surface after the corrosion test showed that A2-0.6Nb steel has a light blue color, but with the aggravation of corrosion, yellow or even yellowish-brown substances began to appear, as shown in Figure 4. When more Fe-rich nodules are formed, vacancies will be left, and it will probably be easier for O ions to enter. Therefore, an IOZ containing Cr and Al was found in the matrix in the substrate inside the oxide film.

The next reason is that adding Nb could prevent the outward diffusion of Cr and Al ions. For instance, we observed a decrease in the amount of Al atoms in the outer oxide film, a trend of inward growth in the inner Al oxide film in A3-2.5Nb steel, and a large Cr- and Al-denuded zone. One explanation is that more ions are consumed for oxidation. Another is that it may also be caused by suppressed outward diffusion. As mentioned above, Shen demonstrated through experiments that the degraded oxidation resistance with high Nb addition mainly resulted from the suppressed outward diffusion of Al atoms by the pinning effect of the Fe_2_Nb phase distributed at boundaries. We are not sure if this effect exists in Fe ions, and generally speaking, the prevention of Cr and Al ions only plays a dominant role after adding excessive Nb.

The effects of laves on oxidation are complicated. On the one hand, Nb as a solid solution can protect the oxide formed in the media to improve the corrosion resistance of stainless steels [46]. Ma et al. [47] found that the laves phase is pinned at grain boundaries and limits grain growth. On the other hand, the laves phase has negative effects on oxidation. Chen [48] recently announced that the oxidation of Fe_2_Nb laves phases at high temperature can lead to cracking and spalling of the oxide layer. The same phenomenon also occurs in other alloys such as Ti-based alloys and ferritic steel [48,49,50,51]. When encountering diffused O ions, Fe_2_Nb is rapidly oxidized to a Nb-O phase, which generates high local stress in the inner oxide layer, leading to the formation of microcracks and pores. Thus, the potential microcracks affect the selective oxidation of AFA reported in different environments.

As a matter of fact, the mechanism of the “model of available space” adequately explains the formation process of cracks and pores. Outer oxides leave more space, whereas the inner oxide layer is not completely filled due to differences in oxide growth rates [32,33]. In addition, sometimes, rapid or insufficient diffusion also leads to significant defects. Uneven pores of all sizes lead to a decrease in the density of the oxide film. Defects such as pores and cracks that occur during the corrosion process are usually detrimental to corrosion, as cracks can easily cause crack propagation, leading to stress corrosion and material degradation [52,53], and the defects in the oxide layer can lead to the formation of more Fe-rich, coarse oxides, thus aggravating corrosion.

Overall, there are several unfavorable factors for oxidation behavior caused by the excessive addition of Nb, including the coarse Fe-rich oxides formed by continuous diffusion through refined grain boundaries, the prevention of the outward diffusion of Al and Cr ions due to precipitates with Nb components, and no more dense and solid oxide film with corrosion defects such as pores and cracks caused by Nb the laves phase.

## 5. Conclusions

The corrosion behavior of three AFA steels with different Nb contents was investigated in the S-CO_2_ environment at 500 °C, 600 °C, and 20 MPa. Based on the results of weight gain, surface oxide, and microstructure analysis of the corroded steels, the following conclusions were drawn.

After corroding in S-CO_2_ for 1000 h, a duplex-oxide layer formed on the surface of AFA steels, which consisted of a uniform and protective outer Cr_2_O_3_ film, dense and solid inner Al_2_O_3_ film, with some discontinuous Fe spinels distributed on the outermost side. Due to the excellent oxidation resistance, the carburization was not observed significantly. Moreover, Cr-C-rich spinels and the γ’-Ni_3_Al phase were produced under the oxide layer. The appearance of the Cr- and Al-denuded zone was caused by the absorption of a large number of elements during the formation of the oxide film, which reflected the degree of oxidation to a certain extent.

As Nb content varies, the chemical composition and structure of the corrosion products do not change fundamentally, only in terms of distribution and quantity. Adding 0.6 wt.% Nb content, the steel had the best corrosion resistance among the three steels. The excellent corrosion performance with the dense oxide film and multi-spinel structures was ascribed to the increasing diffusion of Cr and Al ions through the refined grain boundaries and limiting oxygen ions caused by Nb content.

As a further Nb addition, the corrosion resistance degraded significantly. The outer oxide layer was almost fully covered with coarse Fe-rich spinels with the help of rapid diffusion by refined grains, which led to an external IOZ at the same time. Precipitates with Nb components prevented Al ions from spreading outward and the tendency of inward growth by the pinning effect, and no more dense and solid oxide film with defects such as pores and cracks in the oxidation layer became larger with increasing Nb content, all of which have a negative effect on corrosion behavior.

The weight gains increased significantly along with the thicker oxide film and more precipitates at 600 °C, while it exhibited a more solid and dense oxide film structure at low temperature due to the relatively low degree of diffusion.

## Figures and Tables

**Figure 1 materials-16-04081-f001:**
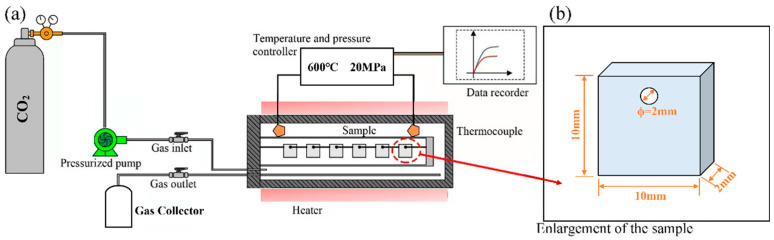
Schematic of S-CO_2_ corrosion device: (**a**) Schematic illustration of the testing system; (**b**) Geometry and dimensions of the specimen.

**Figure 2 materials-16-04081-f002:**
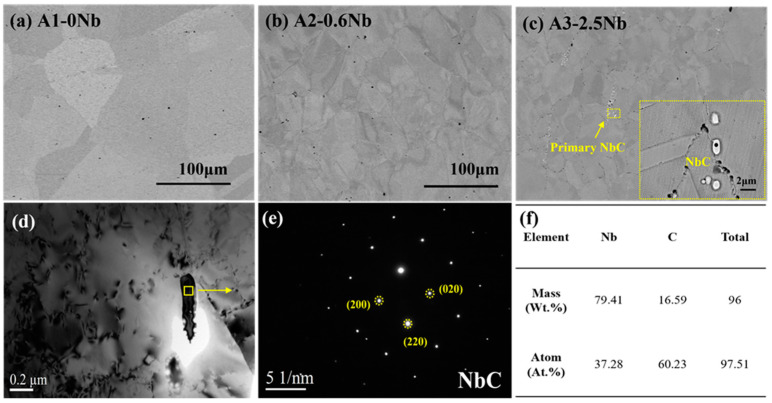
Matrix morphology of AFA steels before corrosion: (**a**–**c**) microstructures of three AFA steels; (**d**,**e**) characteristics of precipitates in AFA steel; (**f**) EDS dot scan result of precipitates.

**Figure 3 materials-16-04081-f003:**
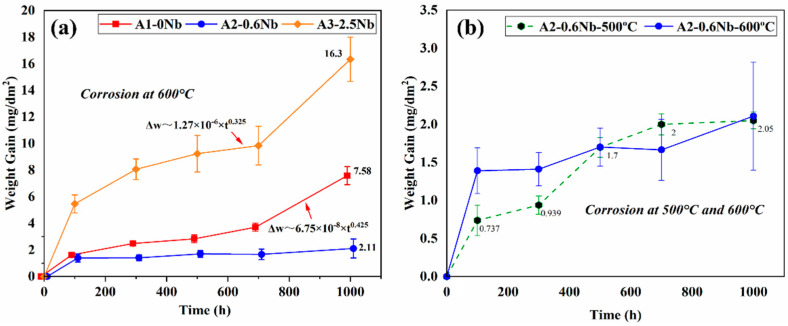
Corrosion kinetics curve of AFA steels after corrosion test in S-CO_2_ environment at 500 °C, 600 °C, and 20 MPa: (**a**) three steels with different Nb contents; (**b**) different temperatures.

**Figure 4 materials-16-04081-f004:**
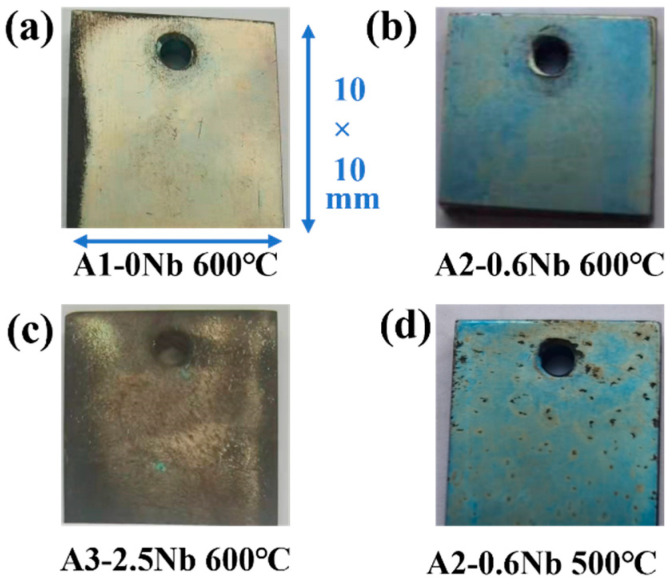
Surface micromorphology of AFA steels after corrosion of 1000 h in S-CO_2_.

**Figure 5 materials-16-04081-f005:**
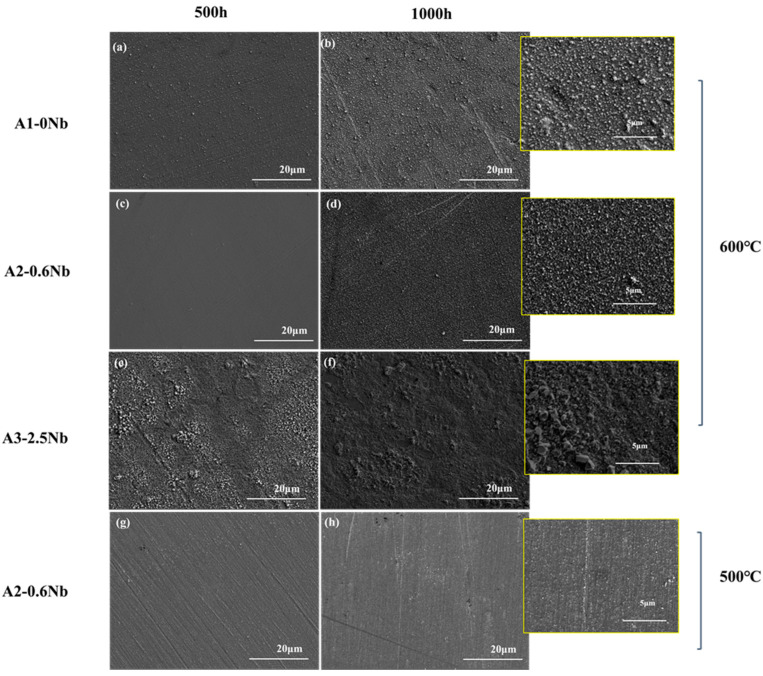
Surface SEM Images for oxide of AFA steels corroded in S-CO_2_ at 500 °C and 600 °C for 500 h and 1000 h: (**a**,**b**) A1 corroded at 600 °C for 500 h and 1000 h; (**c**,**d**) A2 corroded at 600 °C for 500 h and 1000 h; (**e**,**f**) A3 corroded at 600 °C for 500 h and 1000 h; (**g**,**h**) A2 corroded at 500 °C for 500 h and 1000 h.

**Figure 6 materials-16-04081-f006:**
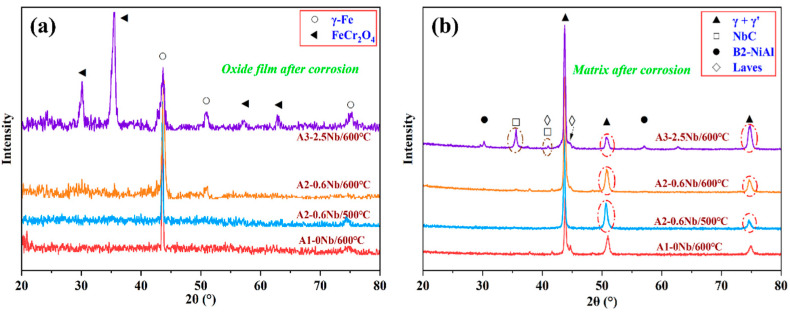
Results of XRD analysis of AFA steels corroded in S-CO_2_ at 500 °C and 600 °C for 1000 h: (**a**) oxide film (PDF#34-0140, 52-0513); (**b**) matrix (PDF#21-0001, 44-1188, 38-1364, 17-0908).

**Figure 7 materials-16-04081-f007:**
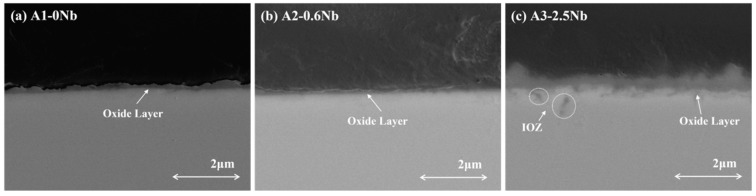
Image of cross-section of AFA steel corroded at 600 °C for 1000 h.

**Figure 8 materials-16-04081-f008:**
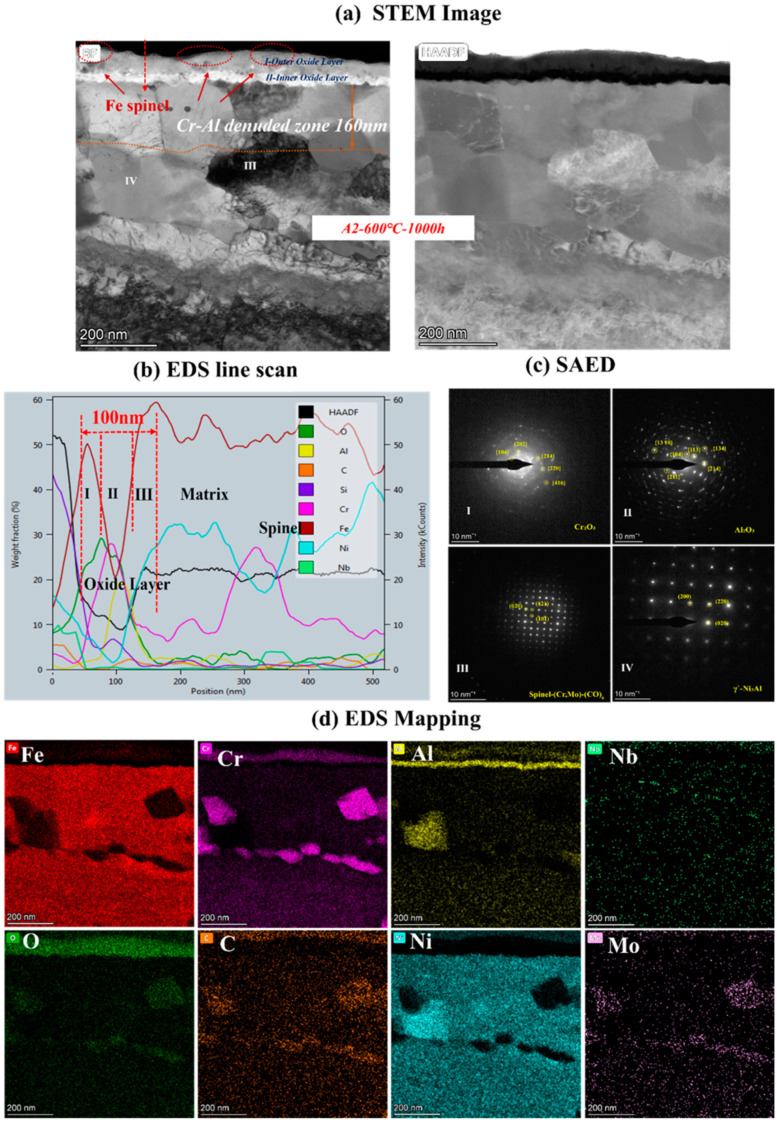
TEM analysis results of A2-0.6Nb steel after exposure to S-CO_2_ at 600 °C for 1000 h; (**a**) cross-sectional micrographs of oxide film; (**b**,**d**) line scanning and EDS mapping results of oxide film; (**c**) phase analysis of oxide film.

**Figure 9 materials-16-04081-f009:**
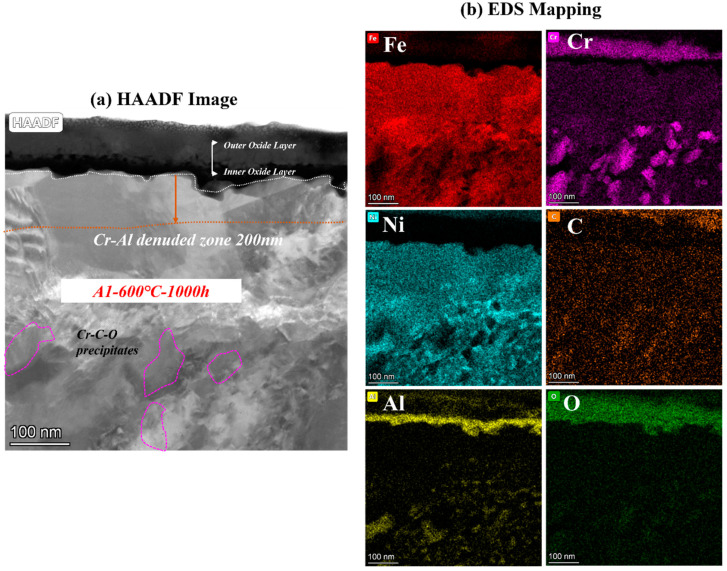
TEM analysis results of A1-0Nb steel after exposure to S-CO_2_ at 600 °C for 1000 h; (**a**) cross-sectional micrographs of oxide film; (**b**) EDS mapping results of oxide film.

**Figure 10 materials-16-04081-f010:**
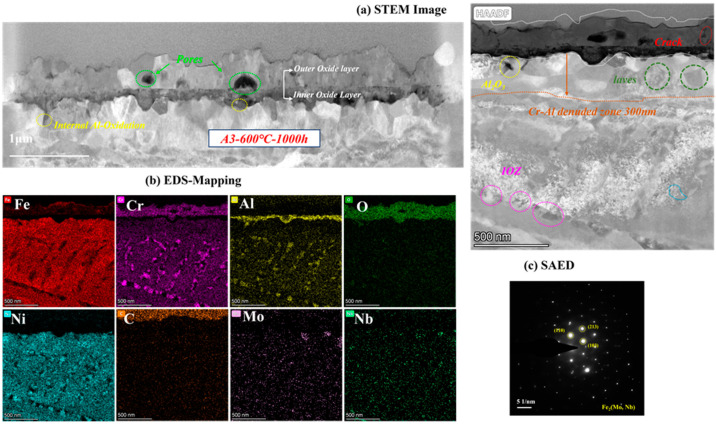
TEM analysis results of A3-2.5Nb steel after exposure to S-CO_2_ at 600 °C for 1000 h; (**a**) cross-sectional micrographs of oxide film; (**b**) EDS mapping results of oxide film; (**c**) phase analysis of precipitate.

**Figure 11 materials-16-04081-f011:**
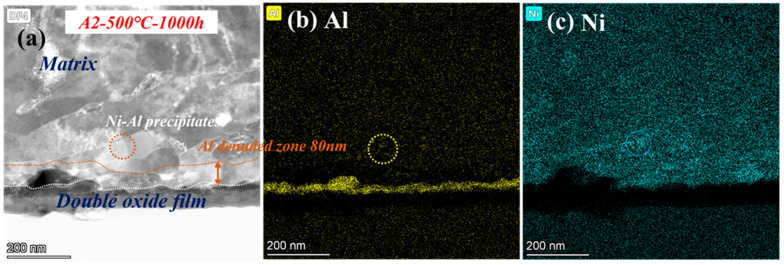
TEM analysis results of A2-0.6Nb steel after exposure to S-CO_2_ at 500 °C for 1000 h; (**a**) cross-sectional micrographs of oxide film; (**b**,**c**) EDS mapping results of oxide film.

**Figure 12 materials-16-04081-f012:**
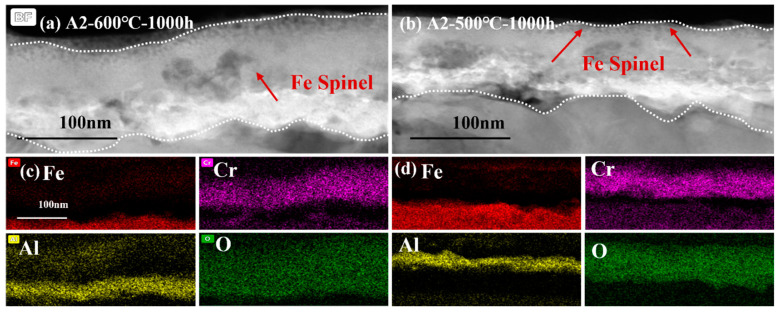
TEM analysis of oxide film of A2-0.6Nb steel corroded in S-CO_2_ at 500 °C and 600 °C for 1000 h: (**a**,**b**) cross-sectional micrographs at 600 °C and 500 °C; (**c**,**d**) EDS mappings at 600 °C and 500 °C.

**Figure 13 materials-16-04081-f013:**
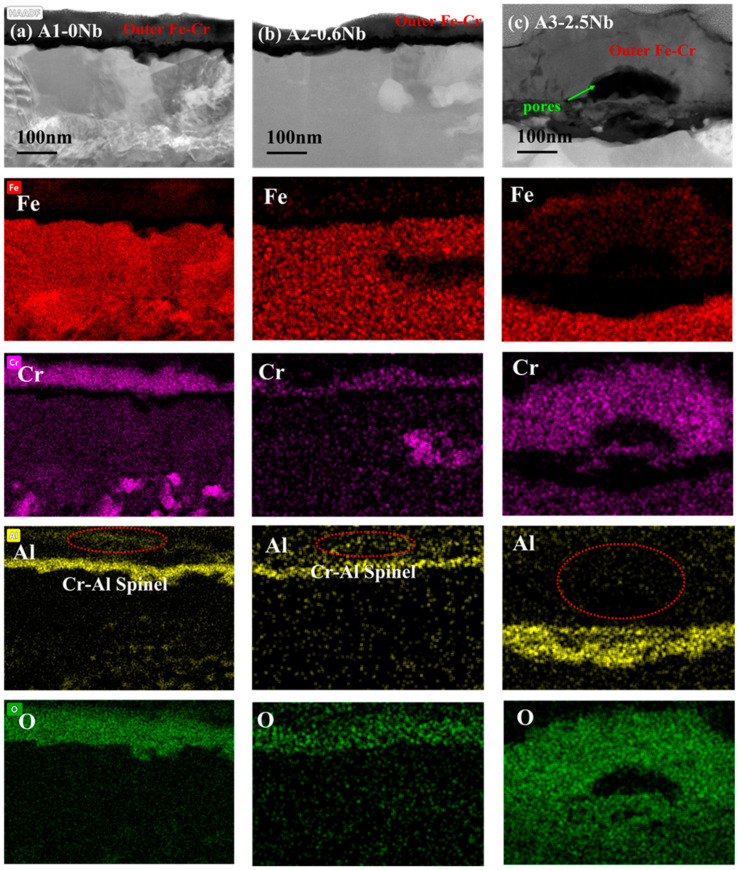
TEM analysis of oxide film of three AFA steels corroded in S-CO_2_ at 600 °C for 1000 h.

**Figure 14 materials-16-04081-f014:**
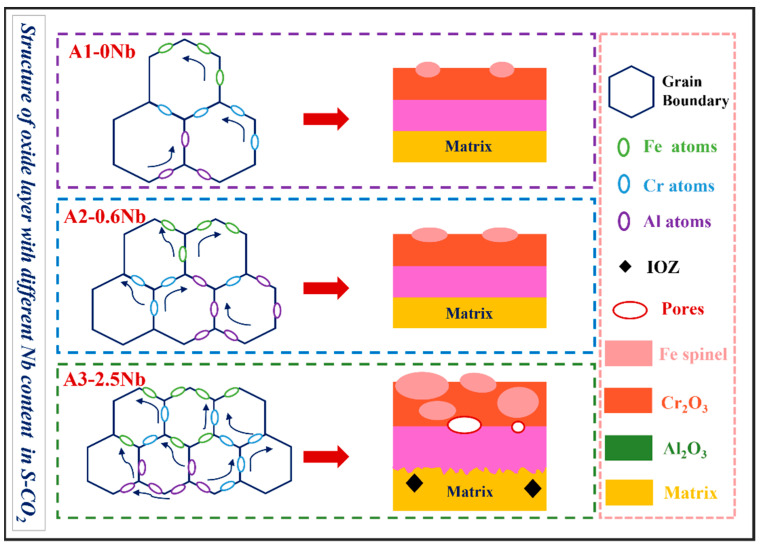
Schematic illustration of corrosion mechanism of three AFA steels with different Nb content.

**Table 1 materials-16-04081-t001:** Normal and actual chemical composition (wt.%) of three AFA steels.

	Materials	Nb	Al	Cr	Ni	Si	Mo	C	Fe
Normal	A1	**0**	3.5	18	25	0.25	2	0.02	Bal.
	A2	**0.6**	3.5	18	25	0.25	2	0.02	Bal.
	A3	**2.5**	3.5	18	25	0.25	2	0.02	Bal.
Actual	A1	**-**	3.7	16.8	24.9	0.3	2.3	0.03	Bal.
	A2	**0.61**	3.6	17.1	25.29	0.3	2.2	0.04	Bal.
	A3	**2.89**	3.7	18.1	23.5	0.3	2.1	0.03	Bal.

## Data Availability

Data available on request due to restrictions e.g., privacy or ethical.

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
