# Peer review of "Corrosion Behavior of Alumina-Forming Austenitic Steel in Supercritical Carbon Dioxide Conditions: Effects of Nb Content and Temperature"

_materials, 2023, doi:10.3390/ma16114081_

Round 1

Reviewer 1 Report

The paper is of interest. However, some changes are required before publication.

-Please add the provider or company of the autoclave

-Please, between number and units introduce a space, i.e.: 100 h or 25 Mpa. Just only with percentages, there is no space (100%)

-Could the authors clarify the change in trend shown in Figure 3(b) between temperature changes?

-The general format of the paper should be reviewed. For example, some figure captions are in “Figure X”, and some “Fig. X”. Review capital letters in the titles of sections and subsections.

-Please use “h” instead of hours (i.e. Lines 181)

-In materials and methods, equipment for TEM analysis and EDS mapping is not explained or defined.

-If Figure 14 is not a result, it should be in materials and methods, as well as reactions.

Reviewer 2 Report

In this work, the authors have studied the corrosion behavior of three alumina-forming steels with different concentrations of Nb. Experiments were carried out in supercritical CO2 at 500 and 600 ° C and pressure 20 MPa. The work is well-researched and sufficiently discussed. The oxide scale was studied by a combination of XRD, SEM, and HR TEM. The paper is worthy of publication subject to revision.

1.The manufacturers and trademarks of the instruments (XRD, SEM, and TEM) should be given in the Materials and Methods section.

2.Since weight gain followed a parabolic kinetics (line 143, Fig. 3), parabolic rate constants should be obtained from the data and compared with previous studies.

3.Fig. 4 should include a scale bar.

4.Images in Fig. 5 should be enlarged. It is advised to present a maximum of 3 images in a row.

5.Powder diffraction file numbers of identified phases should be included in the caption of Fig. 6.

6.Include the names of chemical elements for the EDS maps in Fig. 11.

7.The microstructures in Fig. 13 should be given on the same scale to see the difference in the thickness of the scale clearly. Enlarge Fig. 13c.

8.The schematic in Fig. 14 should clearly distinguish between the elements and their oxides. Use open symbols for the metallic elements located at grain boundaries.

9.An improvement of the language is required. Avoid subjective statements (“As we know”, line 62), grammar mistakes (the structure IS SHOWN (line 261), The effect of Laves PHASE on the oxidation behavior IS TWOFOLD instead of “double-sided”, line 469, etc.), missing words (the AMOUNT of Nd additions increased, line 411), uncertain claims (the word “sometimes” should be avoided, line 469). The paper should be grammar-checked before resubmission.

An improvement of the language is required. Avoid subjective statements (“As we know”, line 62), grammar mistakes (the structure IS SHOWN (line 261), The effect of Laves PHASE on the oxidation behavior IS TWOFOLD instead of “double-sided”, line 469, etc.), missing words (the AMOUNT of Nd additions increased, line 411), uncertain claims (the word “sometimes” should be avoided, line 469). The paper should be grammar-checked before resubmission.

Round 2

Reviewer 2 Report

The authors answered my comments. The paper has been improved. It can be accepted for publication.

A minor English editing is required.